# Genetic associations of *TMEM154*, *PRC1* and *ZFAND6* loci with type 2 diabetes in an endogamous business community of North India

Gagandeep Kaur Walia[1,2]☯*, Pratiksha Sharma[3]☯, Tripti Agarwal[3], Moti Lal[4], Himanshu Negandhi[1], Dorairaj Prabhakaran[1,2], Rajesh Khadgawat[5], Mohinder Pal Sachdeva[4], Vipin Gupta[4]*

1 Public Health Foundation of India, Gurugram, India, 2 Centre for Chronic Disease Control, Safdarjung Development Area, New Delhi, India, 3 Indian Institute of Public Health-Delhi, Public Health Foundation of India, Gurugram, India, 4 Department of Anthropology, University of Delhi, Delhi, India, 5 Department of Endocrinology, All India Institute of Medical Sciences, Ansari Nagar, New Delhi, India

☯ These authors contributed equally to this work.
* gagandeep.k.walia@phfi.org, gkaurw@gmail.com (GKW); drvipiing@gmail.com, udaiig@gmail.com (VG)

**Data Availability Statement:** The data utilized in the present study belongs to the Department of Anthropology, University of Delhi and the de-

## Abstract

### Background

More than 250 loci have been identified by genome-wide scans for type 2 diabetes in different populations. South Asians have a very different manifestation of the diseases and hence role of these loci need to be investigated among Indians with huge burden of cardio-metabolic disorders. Thus the present study aims to validate the recently identified GWAS loci in an endogamous caste population in North India.

### Methods

219 T2D cases and 184 controls were recruited from hospitals and genotyped for 15 GWAS loci of T2D. Regression models adjusted for covariates were run to examine the association for T2D and fasting glucose levels.

### Results

We validated three variants for T2D namely, rs11634397 at *ZFAND6* (OR = 3.05, 95%CI = 1.02–9.19, p = 0.047) and rs8042680 at *PRC1* (OR = 3.67, 95%CI = 1.13–11.93, p = 0.031) showing higher risk and rs6813195 at *TMEM154* (OR = 0.28, 95%CI = 0.09–0.90, p = 0.033) showing protective effect. The combined risk of 9 directionally consistent variants was also found to be significantly associated with T2D (OR = 1.91, 95%CI = 1.18–3.08, p = 0.008). One variant rs10842994 at *KLHDC5* was validated for 9.15mg/dl decreased fasting glucose levels (SE = -17.25–1.05, p = 0.027).

identified data can be requested by sending a collaboration request to the institute at admin@anthro.du.ac.in. The researchers who wish to access this data can mail at the mentioned address requesting Diabetes Genetics Data that belongs to Public Health and Genomics Laboratory (https://anthro.du.ac.in/public%20health.html) of the department. The de-identified data from the present study will be made available without restrictions on collaborator requests. The interested researchers can request by stating the specific research question and a brief analyses plan. The authors of the present study also accessed the data through this route of collaboration. The corresponding authors of the article can also be contacted to facilitate the access of the data, if required.

**Funding:** There was no specific funding for the original study for primary data collection. The laboratory work for genotyping of the variants examined in this article was supported by the UGC-BSR Research Startup grant sanctioned by University Grants Commission (Ref. No. F.30-23/2014(BSR)) to Dr. Vipin Gupta, Department of Anthropology, University of Delhi. The data analyses and the manuscript submission was supported by DBT/Wellcome India Alliance grant (Ref. No. IA/CPHE/16/1/502649) awarded to Dr. Gagandeep Kaur Walia, Public Health Foundation of India.

**Competing interests:** The authors have declared that no competing interests exist.

## Conclusion

We confirm the role of *ZFAND6*, *PRC1* and *TMEM154* in the pathophysiology of type 2 diabetes among Indians. More efforts are needed with larger sample sizes to validate the diabetes GWAS loci in South Asian populations for wider applicability.

## Introduction

Type 2 diabetes (T2D) is a multifactorial chronic disease characterized by hyperglycemia that can lead to circulatory, nervous and immune disorders. It is marked by impaired carbohydrates, lipids, and proteins metabolism caused by complete or partial insufficiency of insulin secretion [1]. The global prevalence of diabetes was reported to be 9.3% in 2019 with higher prevalence in urban (10.8%) than rural (7.2%) areas [2].

Indians are known to be more susceptible to diabetes [3] and that too with younger age of onset and with lower body weights than other populations due to increased metabolic load [4]. With the rapid changes in lifestyle of urban Indians due to high calorie intake and sedentary routine they are likely to suffer from "thin diabetes" making them more vulnerable to metabolic disorders [5]. The prevalence of diabetes in India increased from 5·5% in 1990 to 7·7% in 2016 [6]. The most important risk factor of diabetes in India was reported as overweight that attributed 36% diabetes morbidity in 2016 [6]. For every 100 overweight adults in India, there were 38 diabetics as compared to global average of 19 adults [6]. Lifetime risk of diabetes in adult Indians are alarming with 55% among men and 65% among women and highest proportions among obese individuals [7].

The heritability estimates of T2D range from 30% - 70%, based on twin and family studies in different populations [8–11], providing a compelling evidence about the strong influence of genetic factors. The lifetime risk of developing T2D is 40% for individuals who have one parent with T2D and 70% if both parents are affected [12]. Successive large scale genome-wide association studies (GWASs) have robustly identified ~80 genetic variants in Western populations that are related to diabetes and glycemic levels [13–16]. Till recent years, very limited genome-wide scans were available in Asian populations for T2D [17]. These GWAS findings need to be well validated in Asian populations for generalizability and universal applications. Conducting genetic association studies in India is a challenging task given the high population diversity with over 2000 ethnic groups [18]. There is only one small size GWAS (i.e. 1256 cases and 1209 controls) in relation to diabetes that was conducted in India without recognizing the high genetic diversity and underlying population sub-structure of the collected samples from Delhi [19]. Anthropologically, we must acknowledge and utilize the existing "experiments of nature", i.e. endogamous populations, as the basis of conducting genetic association studies in India.

The identification, validation and characterization, in terms of effect sizes, of disease-informative variants in well-defined population groups are the prime objectives of genetic based association studies. Therefore, the overall aim of the present study is to validate the GWAS loci related to T2D and fasting glucose levels in an endogamous caste population of North India and derive allelic risk scores. These variants were identified in genome-wide scans from western populations and need to be examined in this Indian population with high burden of diabetes.

## Methods

### Study participants and measurements

219 unrelated T2D cases and 184 controls (Table 1), both belonging to Aggarwal caste population, were recruited from out-patient clinic of Maharaja Agrasen Hospital and All Indian

**Table 1. Characteristics of study participants.**

| Characteristics | Cases(n = 219) | Controls(n = 184) | p-value |
|---|---|---|---|
| Mean age (Years) | 57.9(10.5) | 53.8(10.1) | <0.01* |
| Female * | 73/218(33.3) | 89(48.4) | <0.01* |
| Socio economic status * | | | <0.01* |
| Upper | 29/197(14.7) | 15/165(9.09) | |
| Upper Middle | 161/197(81.3) | 147/165(89) | |
| Lower/Lower Middle | 7/197(14.7) | 3/165(1.8) | |
| Physical Activity scale * | | | 0.40 |
| Sedentary | 71/192(36.9) | 51/156(32.7) | |
| Some Physical Activity | 121/192(63.0) | 105/156(67.3) | |
| Body Mass Index (kg/m$^2$) | 28.3(5.2) | 28.5(6.2) | 0.70 |
| Waist Hip Ratio | 1(0.9) | 0.9(0.1) | <0.01* |
| Systolic Blood Pressure (mmHg) | 137.2(18.9) | 129.4(16.3) | <0.01* |
| Diastolic Blood Pressure (mmHg) | 86.9(10.7) | 86.1(10.5) | 0.50 |
| LDL-C (mg/dl) | 105.5(37.2) | 139.8(154.9) | 0.06 |
| HDL-C (mg/dl) | 38.2(7.6) | 41.2(8.1) | 0.01* |
| Triglycerides (mg/dl) | 146.8(91.0) | 110.5(52.4) | <0.01* |
| VLDL-C (mg/dl) | 31.0(21.9) | 22.4(10.4) | <0.01* |
| Total cholesterol(mg/dl) | 173.0(43.9) | 178.6(32.7) | 0.40 |
| Total lipids (mg/dl) | 579.8(133.2) | 539.0(86.3) | 0.03* |
| Cholesterol/HDL-C | 4.6(1.4) | 4.4(0.9) | 0.15 |
| LDL-C/HDL-C | 2.9(1.0) | 2.8(0.8) | 0.71 |

All continuous variables summarized as Mean (SD) and categorical variables (marked as *) as n (%).

*p-value estimated using t-test, ANOVA and Chi square test. Significant difference at p<0.05.

LDL-C = Low Density Lipoprotein-Cholesterol, HDL-C = High Density Lipoprotein-Cholesterol, VLDL-C = Very Low-Density Lipoprotein-Cholesterol.

Institute of Medical Sciences (AIIMS) in New Delhi [20]. This caste population belongs to Vysya community which has been reported as one of the most homogenous groups in India [21,22]. This business community was selected due to high prevalence of diabetes and other metabolic diseases in this population due to their sedentary lifestyle factors. Type 2 Diabetes was defined as fasting glucose levels<126mg/dl and receiving either oral diabetic medication or insulin treatment for previous six months and onset age<30 years. Controls were also recruited from Aggarwal community among individuals >40 years and fasting glucose levels>126mg/dl and not on any kind of diabetes medication. Patients with type 1 diabetes or secondary diabetes due to other reasons were excluded from the study. The original study was ethically cleared from AIIMS, New Delhi and University of Delhi and the present study was approved by Indian Institute of Public Health-Delhi, Public health Foundation of India.

Fasting blood samples were collected from the participants after obtaining written informed consent along with details of their socio-demographic and lifestyle factors. Medical records were gathered for their glycemic and lipid levels. Physical measures included Body Mass Index (BMI), Waist-to-hip ratio (WHR) and blood pressure levels.

## Genotyping

Fifteen Single nucleotide polymorphisms (SNPs) from *SUGP1*, *KLHDC5*, *ZFAND6*, *CHCHD9*, *TLE4*, *HMGA2*, *CENTD2*, *ARAP1*, *FAF1*, *BCL11A*, *ZBED3*, *ZBED3*, *ANK1*, *IRS1*, *HNF1A*, *PRC1* and *KLF14* were selected that were reported as top hits from three large European

**Table 2. Description of genetic variants related to T2D examined in the present study.**

| SNP | Loci | Chr.Position | Alleles | Effect Allele | GWAS Findings | | | EAF$_{Present\ Study}$ | EAF$_{HapMap2}$* | HWE** |
|---|---|---|---|---|---|---|---|---|---|---|
| | | | | | EAF$_{HapMap1}$* | OR(95%CI) | p-value [#] | | | |
| rs10401969 | SUGP1 | 19p13.11 | T/C | C | 0.08 | 1.13(1.09–1.18) | 7.0x10$^{-9}$ [14] | 0.11 | 0.13 | 0.13 |
| rs10842994 | KLHDC5 | 12p11.22 | C/T | C | 0.80 | 1.10(1.06–1.12) | 6.1x10$^{-10}$ [14] | 0.72 | 0.78 | 0.17 |
| rs11634397 | ZFAND6 | 15q25.1 | G/A | G | 0.60 | 1.06(1.04–1.08) | 2.4x10$^{-9}$ [13] | 0.67 | 0.61 | 0.45 |
| rs13292136 | CHCHD9, TLE4 | 9q21.31 | C/T | C | 0.93 | 1.11(1.07,1.15) | 2.8x10$^{-8}$ [13] | 0.14 | 0.09 | 0.02 |
| rs1531343 | HMGA2 | 12q14.3 | G/C | C | 0.10 | 1.10(1.07–1.14) | 3.6x10$^{-9}$ [13] | 0.28 | 0.26 | 0.45 |
| rs1552224 | CENTD2, ARAP1 | 11q13.4 | A/C | A | 0.88 | 1.14(1.11–1.17) | 1.4x10$^{-22}$ [13] | 0.15 | 0.08 | 0.90 |
| rs17106184 | FAF1 | 1p32.3 | G/A | G | 0.90 | 1.10(1.07–1.14) | 4.1x10$^{-9}$ [15] | 0.05 | 0.09 | 0.46 |
| rs243021 | BCL11A | 2p16.1 | A/G | A | 0.46 | 1.08(1.06–1.10) | 2.9x10$^{-15}$ [13] | 0.48 | 0.51 | 0.94 |
| rs4457053 | ZBED3 | 5q13.3 | A/G | G | 0.26 | 1.08(1.06–1.11) | 2.8x10$^{-12}$ [13] | 0.18 | - | 0.01 |
| rs516946 | ANK1 | 8p11.21 | C/T | C | 0.76 | 1.09(1.06–1.12) | 2.5x10$^{-10}$ [14] | 0.75 | 0.71 | 0.72 |
| rs6813195 | TMEM154 | 4q31.3 | C/T | C | 0.72 | 1.08(1.06–1.10) | 4.1x10$^{-14}$ [15] | 0.46 | 0.43 | 0.77 |
| rs7578326 | IRS1 | 2q36.3 | A/G | A | 0.64 | 1.11(1.08–1.13) | 5.4x10$^{-20}$ [13] | 0.16 | 0.29 | 0.72 |
| rs7957197 | HNF1A | 12q24.31 | T/A | T | 0.85 | 1.07(1.05,1.10) | 2.4x10$^{-8}$ [13] | 0.04 | 0.10 | 0.16 |
| rs8042680 | PRC1 | 15q26.1 | A/C | A | 0.22 | 1.07(1.05–1.09) | 2.4x10$^{-10}$ [13] | 0.38 | 0.25 | 0.65 |
| rs972283 | KLF14 | 7q32.2 | G/A | G | 0.55 | 1.07(1.05–1.10) | 2.2x10$^{-10}$ [13] | 0.32 | 0.26 | 0.86 |

EAF = Effect Allele Frequency, OR = Odds Ratio, CI = Confidence Interval.

Effect Allele is the allele showing higher odds of T2D in the original GWAS.

*Effect Allele frequency available from HapMap1 for Caucasians and HapMap2 for South Asians.

** p ≤ 0.003 indicates statistical significance from Pearson chi-square test after Bonferroni Correction for Hardy-Weinberg Equilibrium.

#References of GWAS studies.

studies [13–15]. These genetic variants were selected based on their biological importance and high GWAS significance levels (p<1x10$^{-8}$) related to T2D (Table 2). Since these variants were selected from articles published between 2010–2014, we confirmed their relevance in current NHGRI-GWAS catalogue also for their role in T2D pathophysiology and subsequent studies and meta-analyses also confirm their role in T2D. DNA was isolated from fresh blood samples using salting out method. The genotypes of 15 selected variants were examined by Mass Spectrometry using MassARRAY platform (Agena Biosciences). Intra- and inter- batch duplicates and negative controls were run to assess the data quality.

## Statistical analyses

The analyses were performed in STATA v13.1. Cases and controls were compared for all the variables using chi-square test for categorical variables and t-test for quantitative variables. All the variants were examined for Hardy-Weinberg equilibrium among controls after correcting for multiple testing using Bonferroni correction (Table 2). The effect allele was kept consistent as that reported by original genome-wide studies [13–15]. The allele frequencies of the effect allele of the variants observed in the study population was compared with the HapMap database for South Asians (Table 2). Logistic regression models were run for examining association of each SNP with T2D while assuming additive model of inheritance (Table 3). The regression models were adjusted for covariates in a stepwise manner for age, gender, socio-economic status, physical activity, BMI, WHR, systolic and diastolic blood pressure, and lipid levels. To estimate the combined effect of these GWAS loci, weighted genetic risk score (wGRS) was estimated using all and only directionally consistent variants (same direction of association as seen in original GWAS) in the present study for T2D while weighing for the individual effect

**Table 3. Logistic regression analysis for examining association of genetic variants with Type 2 diabetes.**

| SNP | Loci | OR[a] (95% CI) | p-value | OR[b] (95% CI) | p-value | OR[c] (95% CI) | p-value |
|---|---|---|---|---|---|---|---|
| rs10401969 | SUGP1 | 1.34 (0.70–2.57) | 0.380 | 1.35 (0.68–2.71) | 0.396 | 1.78 (0.35–8.92) | 0.485 |
| rs10842994 | KLHDC5 | 1.08 (0.65–1.81) | 0.760 | 1.12 (0.63–2.01) | 0.693 | 1.69 (0.37–7.66) | 0.496 |
| rs11634397 | ZFAND6 | 1.35 (0.91–2.02) | 0.138 | 1.36 (0.87–2.14) | 0.176 | 3.05 (1.02–9.19) | **0.047** |
| rs13292136 | CHCHD9, TLE4 | 1.34 (0.77–2.35) | 0.302 | 1.29 (0.71–2.34) | 0.400 | 0.31 (0.07–1.38) | 0.125 |
| rs1531343 | HMGA2 | 0.86 (0.55–1.33) | 0.496 | 1.01 (0.62–1.64) | 0.981 | 1.66 (0.46–5.97) | 0.438 |
| rs1552224 | CENTD2, ARAP1 | 1.08 (0.64–1.84) | 0.766 | 1.28 (0.70–2.36) | 0.425 | 0.37 (0.09–1.52) | 0.167 |
| rs17106184 | FAF1 | 0.78 (0.34–1.81) | 0.571 | 0.64 (0.26–1.59) | 0.333 | 0.92 (0.19–4.44) | 0.918 |
| rs243021 | BCL11A | 1.22 (0.84–1.79) | 0.296 | 1.13 (0.74–1.72) | 0.570 | 2.11 (0.71–6.25) | 0.179 |
| rs4457053 | ZBED3 | 1.10 (0.68–1.77) | 0.707 | 0.94 (0.56–1.56) | 0.797 | 0.59 (0.17–2.02) | 0.399 |
| rs516946 | ANK1 | 0.89 (0.53–1.47) | 0.640 | 0.77 (0.44–1.34) | 0.357 | 1.39 (0.34–5.74) | 0.651 |
| rs6813195 | TMEM154 | 0.67 (0.45–1.00) | **0.049** | 0.66 (0.43–1.03) | 0.067 | 0.28 (0.09–0.90) | **0.033** |
| rs7578326 | IRS1 | 1.29 (0.77–2.18) | 0.333 | 1.23 (0.70–2.14) | 0.478 | 1.90 (0.53–6.89) | 0.328 |
| rs7957197 | HNF1A | 0.80 (0.29–2.17) | 0.661 | 0.65 (0.21–2.01) | 0.455 | 0.13 (0.004–4.02) | 0.244 |
| rs8042680 | PRC1 | 1.28 (0.86–1.89) | 0.220 | 1.11 (0.73–1.69) | 0.632 | 3.67 (1.13–11.93) | **0.031** |
| rs972283 | KLF14 | 1.14 (0.76–1.71) | 0.520 | 1.40 (0.89–2.21) | 0.147 | 1.02 (0.35–2.98) | 0.976 |
| wGRS_1 | | | | | | 1.20 (0.88–1.63) | 0.250 |
| wGRS_2 | | | | | | **1.91 (1.18–3.08)** | **0.008** |

[a] Regression Model adjusted for age, gender, SES score, PAS Score, BMI and WHR.

[b] Regression Model adjusted for age, gender, SES score, PAS Score, BMI, WHR, SBP and DBP.

[c] Regression Model adjusted for age, gender, SES score, PAS Score, BMI, WHR, SBP, DBP, HDL-C, LDL-C, VLDL-C, Total cholesterol and Triglycerides.

p-value <0.05 indicates significance.

wGRS_1 –Weighted Genetic Risk Score, of all the 15 genetic variants using external weights from respective GWAS.

wGRS_2 –Weighted Genetic Risk Score, of all the 9 directionally consistent genetic variants (OR>1) using external weights from respective GWAS.

sizes utilized from original studies (Odds Ratio for each variant from original GWAS). The linear regression models were also run for examining the association of these SNPs with fasting blood glucose levels among controls while adjusting for all the covariates (Table 4). As these variants are established GWAS loci for T2D, we did not correct for multiple testing in regression models for the current validation analysis in Indian population.

## Results

The study participants belonged to effluent socio-economic background and overall had high central obesity. Characteristics of the 403 study participants in this study are presented in Table 1. The mean age of cases was 57.9(±10.5) years with 33.3% of females and mean age of controls was 53.8(±10.1) years with 48.4% females. WHR, systolic blood pressure, triglycerides and LDL-C was significantly higher among cases as compared to controls (Table 1). All the variants were found to be in Hardy-Weinberg equilibrium and the effect allele frequency observed in the present study population was in accordance with that reported in HapMap database for South Asians (Table 2). However, the effect allele frequencies varied from the original European studies due to huge ethnic and genomic differences between European and Asian populations.

The cases and controls did not vary significantly for their genotypic distribution and none of the variants were associated with T2D when adjusted for only demographic and lifestyle factors. However, three variants (rs11634397 at ZFAND6, rs8042680 at PRC1 and rs6813195 at TMEM154) were found to be significantly associated with T2D when the regression model was additionally adjusted for lipid levels (Table 3). Each copy of the risk alleles of rs11634397

**Table 4. Linear regression analysis for examining association of genetic variants with fasting glucose levels among controls.**

| SNP | Loci | β[a] (95% CI) | P-value | β[b] (95% CI) | P-value | β[c] (95% CI) | p-value |
|---|---|---|---|---|---|---|---|
| rs10401969 | SUGP1 | -7.55 (-18.43–3.32) | 0.171 | -9.05 (-20.39–2.29) | 0.116 | -9.60 (-31.99–12.80) | 0.387 |
| rs10842994 | KLHDC5 | -8.05 (-15.45 - -0.64) | **0.033** | -9.15 (17.25 - -1.05) | **0.027** | -9.29 (-23.96–5.38) | 0.205 |
| rs11634397 | ZFAND6 | 0.50 (-5.26–6.27) | 0.863 | 0.74 (-5.40–6.88) | 0.810 | -7.75 (-16.56–1.05) | 0.082 |
| rs13292136 | CHCHD9, TLE4 | 7.81 (0.43–15.20) | 0.038 | 7.09 (-0.86–15.05) | 0.080 | 6.37 (-6.84–19.58) | 0.331 |
| rs1531343 | HMGA2 | -4.82 (-11.12–1.48) | 0.132 | -7.10 (-14.12 - -0.09) | 0.047 | -2.20 (-18.52–14.12) | 0.784 |
| rs1552224 | CENTD2, ARAP1 | -4.78 (-12.28–2.71) | 0.208 | -7.01 (-15.02–0.99) | 0.085 | -4.88 (-20.57–10.81) | 0.529 |
| rs17106184 | FAF1 | 9.37 (-7.70–26.45) | 0.278 | 11.14 (-7.38–29.66) | 0.235 | 5.88 (-22.88–34.64) | 0.678 |
| rs243021 | BCL11A | 0.95 (-4.66–6.57) | 0.736 | 0.25 (-6.09–6.60) | 0.938 | -1.73 (-12.09–8.63) | 0.734 |
| rs4457053 | ZBED3 | 1.52 (-5.43–8.46) | 0.665 | 1.47 (-5.91–8.85) | 0.693 | 8.03 (-4.82–20.87) | 0.211 |
| rs516946 | ANK1 | 1.23 (-6.69–9.14) | 0.759 | 2.21 (-6.33–10.74) | 0.609 | -2.67 (-15.76–10.43) | 0.679 |
| rs6813195 | TMEM154 | 0.01 (-6.33–6.35) | 0.998 | 0.69 (-5.94–7.31) | 0.837 | 6.56 (-5.14–18.27) | 0.260 |
| rs7578326 | IRS1 | -0.03 (-7.88–7.83) | 0.995 | 0.01 (-8.50–8.52) | 0.998 | -7.91 (-22.21–6.38) | 0.266 |
| rs7957197 | HNF1A | 5.98 (-8.10–20.07) | 0.401 | 7.80 (-7.69–23.28) | 0.319 | 0.54 (-43.73–44.82) | 0.980 |
| rs8042680 | PRC1 | 0.94 (-5.28–7.15) | 0.765 | 1.78 (-4.98–8.54) | 0.602 | -6.70 (-17.07–3.67) | 0.196 |
| rs972283 | KLF14 | -5.39 (-11.83–1.04) | 0.099 | -5.01 (-11.95–1.94) | 0.155 | -9.37 (-21.03–2.29) | 0.111 |

[a] Regression Model adjusted for age, gender, SES score, PAS Score, BMI and WHR.

[b] Regression Model adjusted for age, gender, SES score, PAS Score, BMI, WHR, SBP and DBP.

[c] Regression Model adjusted for age, gender, SES score, PAS Score, BMI, WHR, SBP, DBP, HDL-C, LDL-C, VLDL-C, Total cholesterol and Triglycerides.

p-value <0.05 indicates significance.

at *ZFAND6* and rs8042680 at *PRC1* were associated with more than 3-fold increased odds of having T2D (OR = 3.05, 95%CI = 1.02–9.19, p = 0.047 and OR = 3.67, 95%CI = 1.13–11.93, p = 0.031, respectively). On the other hand, the effect allele of rs6813195 at *TMEM154* showed protective effect (OR = 0.28, 95%CI = 0.09–0.90, p = 0.033) in the present study. The combined effect of all the variants together was not found to be statistically significant but the combined effect of all the directionally consistent 9 variants (OR>1, using same effect allele as reported by original GWAS) showed significant increased odds of having T2D (OR = 1.91, 95% CI = 1.18–3.08, p = 0.008). In addition, one more variant was found to be significantly associated with fasting blood glucose levels among control population after adjusting for all demographic and lifestyle factors. Each copy of the C allele of rs10842994 at *KLHDC5* gene was associated with -9.15 mg/dl (SE = -17.25 - -1.05, p = 0.027) decreased fasting glucose levels.

## Discussion

The overall aim of the study was to validate the GWAS loci of T2D among an endogamous population of North India. We have found associations of genetic variants on/nearby *ZFAND6 and PRC1* loci with T2D in the studied 'Aggarwal' population with OR>3 for each variant and *TMEM154* loci showing protective effect for T2D. Not all the variants showed directionally consistent effects as that reported in the original genome-wide studies conducted among European populations [13–15]. Some of the variants showed protective effect for T2D while taking same allele as the risk or effect allele as reported by the original studies [13–15]. Thus, in order to estimate the combined effect of different variants, we considered only directionally consistent variants. The weighted genetic risk score of the 9 directionally consistent variants was associated with 1.9-fold higher odds of developing T2D. Further, it needs to be noted that these variants were found to be significantly associated with T2D without adjusting for multiple testing. As these GWAS loci were established in large scale studies and well replicated at a

very low significance value, we did not account for multiple testing in the current validation analyses. None of these variants are significantly associated with T2D in current population after correcting for multiple testing (corrected p<0.0033).

These loci exhibited much lower effect sizes of OR<1.2 among European populations [13–15]. Comparatively higher effect sizes for each of the associated loci in presently studied population is due to careful selection of the endogamous population. The selection of endogamous population provided a defined genetic pool reducing the bias due to population stratification. Previously we had validated *TCF7L2*, *HHEX*, *KCNJ11*, and *ADIPOQ* loci for T2D in a large multi-centric Indian setting in 2010 [20]. We have now also validated the role of *KLHDC5* in regulating the fasting glucose levels. The sedentary lifestyle and high calorie intake of this ancestrally business community urges the need for understanding the diseases etiology in this particular population. The effect modification and gene-environment interaction analyses in this population requiring larger samples is required to further investigate the role of lifestyle factors. Difference in effect sizes and especially the difference in direction of the effect of some of the variants observed in the present study could also be because of underlying genetic heterogeneity and interaction with a very different lifestyle and environmental factors between European and Indian settings. The socio-cultural practices play a huge role in gene-environment interactions that are very distinct and diverse in India with a lot of geographic differences as well. Our emphasis on reducing the population heterogeneity while selecting the populations can also be observed in other large studies, like among American Indians [23] and Danish population [24], where they had observed the best replication for *TMEM154* (rs6813195) on chromosome-4 with T2D. The GWAS with limited sample size conducted in India had found strongest association of a newly identified locus [i.e. 2q21] nearby *TMEM163* on chromosome-2 with TDM which may be due to underlying population stratification [19]. The study suggested a plausible effect of this region through impaired insulin secretion in T2D etiology [19].

Although *ZFAND6* and *PRC1* were identified as diabetes related loci in western populations [13], only recently the role of *ZFAND6* and *PRC1* loci have been described in insulin secretion and beta cell function through animal models [25,26]. The *PRC1* locus is known for encoding protein convertase enzyme, i.e. FURIN, that initiates proteolytic activation of several precursor proteins within the secretory pathway of beta cells [27]. This variant is not well examined for its role in diabetes and glycemic traits in other populations. The genetic variants in *ZFAND3* has shown association with T2D in East Asian population also [28]. Studies conducted among Chinese population has indicated that carrying T2D risk allele G of rs11634397 near *ZFAND6* showed association with the increased risk for elevated triglyceride levels which indicates that *ZFAND6* has a potential role in lipid metabolism also [29]. Further, the transcriptomic studies have shown that it disrupts islet enhancer activity by altering sequence-specific DNA binding of NEUROD1 which is an islet-enriched transcription factor [30].

European studies suggest that C allele of rs6813195 at *TMEM154* is linked to insulin sensitivity and secretion [24]. Very few studies have examined the genetic determinants of surrogate markers of beta cell function and insulin sensitivity. This variant in *TMEM154* has been associated with disposition index in smaller cohorts and has shown moderate connection with a lower insulinogenic index in meta-analyses [24]. A reduced disposition index value could be caused by continuous exposure to insulin resistance triggers (e.g. high calorie diet or sedentary lifestyle) without beta cell compensation, resulting in the progression to pre diabetes. The insulinogenic index is a ratio that compares the amount of circulating insulin augmentation to the magnitude of the associated glycemic stimulus, and it is thus a measure of early insulin responsiveness. These previously published findings give credibility to the hypothesis that rs6813195 at *TMEM154* has a deleterious impact on beta cell function. However, the protective effect for

T2D exhibited by C allele in the present study needs is in consistent with studies conducted among South Asian populations [31]. This strongly supports our idea that findings from European populations need to be validated in different populations using endogamous populations because of genetic heterogeneity and complex gene-environment interactions. Further, we could validate only three out of fifteen variants for T2D that were examined in the present study with limited sample size due to inclusion of only one endogamous population. These variants have not been in exclusively studied in India but their role in T2D has been confirmed in other Asian populations [32–36]. Therefore, there could be false negative findings in our study that need to be interpreted carefully.

To conclude, we confirm the role of *ZFAND6*, *PRC1* and *TMEM154* loci in the etiology of T2D in an endogamous caste population in North India. More such attempts are needed in other Indian populations to understand the unique diabetes manifestation among Indians. Further studies involving larger samples are required to understand the role of lifestyle factors in modifying the genetic effects.

## Supporting information

**S1 File.**
(DOCX)

## Author Contributions

**Conceptualization:** Gagandeep Kaur Walia, Vipin Gupta.

**Data curation:** Pratiksha Sharma, Moti Lal, Rajesh Khadgawat, Vipin Gupta.

**Formal analysis:** Pratiksha Sharma, Tripti Agarwal.

**Funding acquisition:** Gagandeep Kaur Walia, Vipin Gupta.

**Investigation:** Pratiksha Sharma, Moti Lal, Vipin Gupta.

**Methodology:** Gagandeep Kaur Walia, Pratiksha Sharma, Tripti Agarwal, Rajesh Khadgawat, Vipin Gupta.

**Project administration:** Gagandeep Kaur Walia, Vipin Gupta.

**Resources:** Gagandeep Kaur Walia, Dorairaj Prabhakaran, Rajesh Khadgawat, Mohinder Pal Sachdeva, Vipin Gupta.

**Software:** Pratiksha Sharma.

**Supervision:** Gagandeep Kaur Walia, Tripti Agarwal, Himanshu Negandhi, Dorairaj Prabhakaran, Rajesh Khadgawat, Mohinder Pal Sachdeva, Vipin Gupta.

**Validation:** Gagandeep Kaur Walia, Vipin Gupta.

**Writing – original draft:** Gagandeep Kaur Walia, Pratiksha Sharma.

**Writing – review & editing:** Gagandeep Kaur Walia, Pratiksha Sharma, Tripti Agarwal, Moti Lal, Himanshu Negandhi, Dorairaj Prabhakaran, Rajesh Khadgawat, Mohinder Pal Sachdeva, Vipin Gupta.

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
