## [Decision Letter · Decision Letter 0]

30 Mar 2023

PONE-D-23-06568Genetic Associations of TMEM154, PRC1 and ZFAND6 loci with Type 2 Diabetes in an Endogamous Business Community of North IndiaPLOS ONE

Dear Dr. Walia,

Thank you for submitting your manuscript to PLOS ONE. After careful consideration, we feel that it has merit but does not fully meet PLOS ONE’s publication criteria as it currently stands. Therefore, we invite you to submit a revised version of the manuscript that addresses the points raised by the reviewers.

We look forward to receiving your revised manuscript.

Kind regards,

Yun Li

Academic Editor

PLOS ONE

Journal Requirements:

Reviewers' comments:

Reviewer's Responses to Questions

**Comments to the Author**

1. Is the manuscript technically sound, and do the data support the conclusions?

Reviewer #1: Partly

Reviewer #2: Yes

2. Has the statistical analysis been performed appropriately and rigorously? 

Reviewer #1: Yes

Reviewer #2: Yes

3. Have the authors made all data underlying the findings in their manuscript fully available?

Reviewer #1: No

Reviewer #2: No

4. Is the manuscript presented in an intelligible fashion and written in standard English?

Reviewer #1: Yes

Reviewer #2: Yes

5. Review Comments to the Author

Reviewer #1: In this manuscript, the authors assessed the genetic associations of selected variants with Type 2 Diabetes (T2D) in an Indian community. Overall, the manuscript is easy to follow, but I have several concerns regarding the validity and generalizability of the study.

Major comments:

1. It is unclear to me how these variants were selected. The authors only mentioned that “variants were selected based on their biological importance and high GWAS significance levels related to T2D”, but details are lacking regarding the significance threshold and the rationale of these specific variants.

2. The authors referred to references 13 and 15 for selecting variants, which were published in 2010 and 2012 respectively, which were outdated. Are these variants show any significant associations with T2D in more recently published GWAS? For example, the 2022 Nature Genetics T2D study (PMID: 35551307).

3. In the allelic risk score analysis, it seems the authors calculated scores based on the effects estimated from all the participants, and then tested the associations of the scores in the same populations, which will result in severe overfitting issue of the scores. To ensure such scores were valid, I suggest the authors split all the participants into training and testing parts, derive the weights from training individuals and test for the associations in the testing individuals.

4. The reason that the authors didn’t perform any multiple testing adjustment is not convincing to me. Alternatively, the authors could test all the variants in one model due to the small number of variants included in this study.

Other comments:

1. Define “wGRS” in abstract.

2. Define “DALYs” in the second paragraph of the abstract.

3. Lifestyle, gene-environment interactions could also be a source of heterogeneity. Some discussions will be helpful.

Reviewer #2: In this manuscript, the authors aimed to assess whether T2D-GWAS loci originally identified in European populations could also contribute to T2D susceptibility among the North Indian population. To accomplish this, they selected 15 GWAS loci previously identified in European populations from two prior studies. Genotype data from North Indian individuals were then measured and analyzed for these 15 loci to investigate the association between the loci and T2D, as well as T2D-related traits. The results revealed that out of the 15 loci, 3 showed nominal associations (p-value < 0.05) in North Indian individuals. I have a few comments regarding this manuscript. A detailed list of comments can be found below.

List of comments:

1. The authors noted that they did not apply multiple testing adjustment to the SNP association results since the SNPs were identified from European ancestry. And the genetic association for TMEM154, PRC1, and ZFAND6 loci were validated based on nominal p-values. However, none of the SNPs are significantly associated with the outcome after applying the Bonferroni correction. The authors could discuss the Bonferroni-corrected results to provide a more comprehensive assessment of the genetic associations observed in the study.

2. The authors compared the results in terms of risk alleles and major alleles between the regression results and prior literature results. However, it can be somewhat confusing to interpret the data presented in Table 3 since it is unclear which allele is the risk allele. To address this, it would be helpful if the authors could include the risk allele in European ancestry from the two prior studies (references (13) and (15)) as well as the risk allele used in the regression analyses in this manuscript. Additionally, the authors should clarify the inconsistency between the p-values based on minor alleles and p-values based on major alleles for rs11634397 and rs8042680.

3. The authors discussed the directions of odds ratios between prior GWAS results in European studies and regression results in Table 3. To further clarify the consistency of the findings, it would be beneficial if the authors could compare the directions of odds ratios for all 15 SNPs with prior GWAS results. This would enable readers to better assess the potential impact of population differences on the genetic associations.

4. The authors reported the results of a weighted genetic risk score (wGRS) constructed using three significant SNPs identified in the Indian population. It may not fully demonstrate the potential of the 15 SNPs’ contribution to T2D-susceptibility in the Indian population. Therefore, it would be useful if the authors could construct a wGRS using all 15 SNPs to investigate whether individuals with high risk scores have significantly larger odds ratios than those with low risk scores.

5. Given the limited power of this study, some of the non-significant SNPs among the 15 SNPs tested may represent false negatives. To address this possibility, it would be informative if the authors could provide evidence of association for these non-significant SNPs in other South Asian studies.

6. In order to further validate the 15 SNPs identified from European populations, it could be helpful for the authors to explore potential interactions with other risk factors for T2D, such as conducting genotype-by-BMI interaction analysis for the three significant SNPs, as well as wGRS-by-covariate interaction analysis. Additionally, a meta-analysis of prior South Asian GWAS results could be conducted to strengthen the validation of these 15 SNPs.

7. In the Introduction section, the authors cited the paper authored by Kooner et al. (reference (14)) when discussing GWAS analysis of T2D in western populations. However, this paper should not be cited here, as Kooner et al.’s work is focused on South Asian ancestry. This work should be discussed in other parts of this manuscript.

8. In Table 2, it would be beneficial for the authors to provide additional information about the SNP, such as its position and chromosome number.

6. PLOS authors have the option to publish the peer review history of their article (what does this mean?). If published, this will include your full peer review and any attached files.

Reviewer #1: No

Reviewer #2: No

---

## [Author Response · Author response to Decision Letter 0]

8 Jun 2023

We would like to thank the editorial board for taking forward our manuscript and for providing the valuable reviewer’s feedback that has helped in improving the manuscript. We have now revised the manuscript according the reviewer comments and have provided below the response to each of the comment.

Reviewer #1: In this manuscript, the authors assessed the genetic associations of selected variants with Type 2 Diabetes (T2D) in an Indian community. Overall, the manuscript is easy to follow, but I have several concerns regarding the validity and generalizability of the study.

Major comments:

1. It is unclear to me how these variants were selected. The authors only mentioned that “variants were selected based on their biological importance and high GWAS significance levels related to T2D”, but details are lacking regarding the significance threshold and the rationale of these specific variants.

Response: We would like to thank the reviewer for raising this concern. We agree that the rationale for selecting the studied genetic variants should be detailed in the methods section. We have now provided these details in the revised manuscript and also in the SNP description table no. 2.

2. The authors referred to references 13 and 15 for selecting variants, which were published in 2010 and 2012 respectively, which were outdated. Are these variants show any significant associations with T2D in more recently published GWAS? For example, the 2022 Nature Genetics T2D study (PMID: 35551307).

Response: Thanks for providing the suggestion for examining the relevance of previously reported GWAS loci in more recent context. We have checked the recent GWAS catalogue for T2D related genetic variants and confirm the role of the presently examined genetic variants in the pathophysiology of T2D. We also cross-checked the GWAS catalogue for subsequent GWAS and meta-analyses that confirms their relevance in T2D. We have mentioned this in the revised manuscript also.

3. In the allelic risk score analysis, it seems the authors calculated scores based on the effects estimated from all the participants, and then tested the associations of the scores in the same populations, which will result in severe overfitting issue of the scores. To ensure such scores were valid, I suggest the authors split all the participants into training and testing parts, derive the weights from training individuals and test for the associations in the testing individuals.

Response: We agree with the reviewer that the use of internal weights for the combined genetic risk score can introduce the bias. However, we could not divide the data into training and testing parts as suggested by the reviewer due to low sample size. Instead, we have now used the external weights from the literature as suggested by the other reviewer and is well-accepted method for deriving weighted genetic risk scores. 

4. The reason that the authors didn’t perform any multiple testing adjustment is not convincing to me. Alternatively, the authors could test all the variants in one model due to the small number of variants included in this study.

Response: We agree that it is matter of debate to adjust / not adjust for multiple testing while validating the already established GWAS loci. Both type of articles can be found in the literature. As suggested by the reviewer, we examined all the variants in the same model but the results varied a lot from the single SNP association results. Therefore, for careful interpretation, we have mentioned in the discussion section that these associations are not significant after multiple testing correction. 

Other comments:

1. Define “wGRS” in abstract.

Response: We have now revised the text in the abstract accordingly.

2. Define “DALYs” in the second paragraph of the abstract.

Response: We have now edited text accordingly to avoid using the abbreviation. 

3. Lifestyle, gene-environment interactions could also be a source of heterogeneity. Some discussions will be helpful.

Response: We have included this in the revised manuscript in the discussion section.

Reviewer #2: In this manuscript, the authors aimed to assess whether T2D-GWAS loci originally identified in European populations could also contribute to T2D susceptibility among the North Indian population. To accomplish this, they selected 15 GWAS loci previously identified in European populations from two prior studies. Genotype data from North Indian individuals were then measured and analyzed for these 15 loci to investigate the association between the loci and T2D, as well as T2D-related traits. The results revealed that out of the 15 loci, 3 showed nominal associations (p-value < 0.05) in North Indian individuals. I have a few comments regarding this manuscript. A detailed list of comments can be found below.

List of comments:

1. The authors noted that they did not apply multiple testing adjustment to the SNP association results since the SNPs were identified from European ancestry. And the genetic association for TMEM154, PRC1, and ZFAND6 loci were validated based on nominal p-values. However, none of the SNPs are significantly associated with the outcome after applying the Bonferroni correction. The authors could discuss the Bonferroni-corrected results to provide a more comprehensive assessment of the genetic associations observed in the study.

Response: We agree that it is matter of debate to adjust / not adjust for multiple testing while validating the already established GWAS loci. Both type of articles can be found in the literature. We have now mentioned the findings after multiple testing correction also in the revised manuscript under the discussion section for careful interpretation. 

2. The authors compared the results in terms of risk alleles and major alleles between the regression results and prior literature results. However, it can be somewhat confusing to interpret the data presented in Table 3 since it is unclear which allele is the risk allele. To address this, it would be helpful if the authors could include the risk allele in European ancestry from the two prior studies (references (13) and (15)) as well as the risk allele used in the regression analyses in this manuscript. Additionally, the authors should clarify the inconsistency between the p-values based on minor alleles and p-values based on major alleles for rs11634397 and rs8042680.

Response: Thanks for raising this issue. We have now clearly addressed this in the revised manuscript both in methods and discussion sections. We have cross-checked and ensured that the effect allele is same in our study and the original GWAS and have clarified this in the table-2 also. All the association analyses were re-run by taking same effect allele as in the original GAWS study. 

3. The authors discussed the directions of odds ratios between prior GWAS results in European studies and regression results in Table 3. To further clarify the consistency of the findings, it would be beneficial if the authors could compare the directions of odds ratios for all 15 SNPs with prior GWAS results. This would enable readers to better assess the potential impact of population differences on the genetic associations.

Response: Thanks for pointing out the importance of directional consistency of the association results. We have compared the directional consistency of the variants with the previous findings and have discussed the same in the revised manuscript. We have also re-generated the combined wGRS for only the directionally consistent 9 variants. Taking the example of TMEM154, we have also discussed the direction of effect can differ between European and Asian populations. 

4. The authors reported the results of a weighted genetic risk score (wGRS) constructed using three significant SNPs identified in the Indian population. It may not fully demonstrate the potential of the 15 SNPs’ contribution to T2D-susceptibility in the Indian population. Therefore, it would be useful if the authors could construct a wGRS using all 15 SNPs to investigate whether individuals with high risk scores have significantly larger odds ratios than those with low risk scores.

Response: We have now constructed the wGRS using all the examined variants using external weights from the original studies, and also for only the 9 directionally consistent variants using external weights. The results are presented in the table no 3 in the revised manuscript.

5. Given the limited power of this study, some of the non-significant SNPs among the 15 SNPs tested may represent false negatives. To address this possibility, it would be informative if the authors could provide evidence of association for these non-significant SNPs in other South Asian studies.

Response: Thanks for raising this concern of false negative findings. We have addressed this limitation in the discussion section in the revised manuscript. We reviewed literature for these variants among South Asian populations and have discussed the same in the revised manuscript.

6. In order to further validate the 15 SNPs identified from European populations, it could be helpful for the authors to explore potential interactions with other risk factors for T2D, such as conducting genotype-by-BMI interaction analysis for the three significant SNPs, as well as wGRS-by-covariate interaction analysis. Additionally, a meta-analysis of prior South Asian GWAS results could be conducted to strengthen the validation of these 15 SNPs.

Response: We had explored the effect modification for a range of related risk factors, but did not find any significant gene-environment interaction due to limited sample size. We agree that a meta-analysis of south Asian GWAS findings can be very insightful. However, meta-analysis is beyond the scope of this manuscript due to limited resources at present. We will definitely try to do this in future.

7. In the Introduction section, the authors cited the paper authored by Kooner et al. (reference (14)) when discussing GWAS analysis of T2D in western populations. However, this paper should not be cited here, as Kooner et al.’s work is focused on South Asian ancestry. This work should be discussed in other parts of this manuscript.

Response: Thanks for pointing out this error. We have revised the manuscript accordingly.

8. In Table 2, it would be beneficial for the authors to provide additional information about the SNP, such as its position and chromosome number.

Response: We have now provided all the details of the variants in the table 2 in the revised manuscript.

---

## [Decision Letter · Decision Letter 1]

29 Aug 2023

Genetic Associations of TMEM154, PRC1 and ZFAND6 loci with Type 2 Diabetes in an Endogamous Business Community of North India

PONE-D-23-06568R1

Dear Dr. Walia,

We’re pleased to inform you that your manuscript has been judged scientifically suitable for publication and will be formally accepted for publication once it meets all outstanding technical requirements.

Kind regards,

Narasimha Reddy Parine, Ph.D

Academic Editor

PLOS ONE

Reviewers' comments:

Reviewer's Responses to Questions

**Comments to the Author**

1. If the authors have adequately addressed your comments raised in a previous round of review and you feel that this manuscript is now acceptable for publication, you may indicate that here to bypass the “Comments to the Author” section, enter your conflict of interest statement in the “Confidential to Editor” section, and submit your "Accept" recommendation.

Reviewer #1: All comments have been addressed

Reviewer #2: All comments have been addressed

2. Is the manuscript technically sound, and do the data support the conclusions?

Reviewer #1: Yes

Reviewer #2: (No Response)

3. Has the statistical analysis been performed appropriately and rigorously? 

Reviewer #1: Yes

Reviewer #2: (No Response)

4. Have the authors made all data underlying the findings in their manuscript fully available?

Reviewer #1: No

Reviewer #2: (No Response)

5. Is the manuscript presented in an intelligible fashion and written in standard English?

Reviewer #1: Yes

Reviewer #2: (No Response)

6. Review Comments to the Author

Reviewer #1: I would like to thank the authors for revising the manuscript. All my comments have been addressed and I have no further comments.

Reviewer #2: (No Response)

7. PLOS authors have the option to publish the peer review history of their article (what does this mean?). If published, this will include your full peer review and any attached files.

Reviewer #1: No

Reviewer #2: No

---

## [Editor Report · Acceptance letter]

14 Sep 2023

PONE-D-23-06568R1 

Genetic Associations of *TMEM154, PRC1 and ZFAND6* loci with Type 2 Diabetes
in an Endogamous Business Community of North India 

Dear Dr. Walia:

I'm pleased to inform you that your manuscript has been deemed suitable for publication in PLOS ONE. Congratulations! Your manuscript is now with our production department. 

Kind regards, 

on behalf of

Dr. Narasimha Reddy Parine 

Academic Editor

PLOS ONE